

# Difference of SPAC composition and control factors of different vegetation zones in north slope of Qilian Mountains

Yuwei Liu[a,b], Guofeng Zhu[a,b,*], Zhuanxia Zhang[a,b], Zhigang Sun[a,b], Leilei Yong [a,b], Liyuan Sang[a,b], Lei Wang[a,b], Kailiang Zhao[a,b]

a School of Geography and Environment Science, Northwest Normal University, Lanzhou 730070, Gansu, China

b Shiyang River Ecological Environment Observation Station, Northwest Normal University, Lanzhou 730070, Gansu, China

*Correspondence to*: Guofeng Zhu (_gfzhu@lzb.ac.cn)

**Abstract:** Understanding the differences and controlling factors of stable water isotopes in the soil-plant-atmosphere continuum of different vegetation zones has important guiding significance for revealing the hydrological processes and regional water cycle mechanisms.This study selected three different vegetation zones (alpine meadows, forests, and arid piedmont zones) in the Shiyang River Basin for study. This paper's analysis results show that: (1) In SPAC, precipitation isotope has the main controlling effect. From alpine meadows to arid foothills, as the altitude decreases, the temperature effect of precipitation isotopes increases. (2) From the alpine meadow to the arid foothills, the soil water isotope is gradually enriched, indicating that the evaporation is gradually increasing. (3) Alpine meadow plants are mainly supplied by precipitation in the rainy season; forest plants mainly utilize soil water in the dry season and precipitation in the rainy season. The soil water in the arid mountain foothills is mainly recharged by groundwater, and the evaporation and fractionation of plant isotopes are very strong.This research will help understand the SPAC system's water cycle at different altitudes and climates on high mountains.

**Keywords**: Shiyang River Basin; Stable isotope; Precipitation; Soil water; Plant water

**1 Introduction**

The relative abundance changes of isotope technology in water can indicate the water cycle and water use mechanism in plants, so isotope technology has become an increasingly important method for studying the water cycle (Gao et al. 2009; Song et al. 2002; Coplen, 2013; Shou et al. 2013). The





stable isotope composition of water is considered to be the "fingerprint" of water, which records a large
amount of environmental information that comprehensively reflects the geochemical process of each
system and links the composition characteristics of each link (Darling et al., 2003; Rac et al., 2013; Gaj
et al., 2014; Nlend et al., 2020). Moreover, it is used in the research of the analysis of water sources,
migration and mixing, and other dynamic processes and played an increasingly important role (White
et al., 2013; Bowen et al., 2015). In particular, D and 18O are considered conservative and stable in the
absence of high-temperature water-rock interaction and strong evaporation conditions. They are the
ideal environmental isotopes for tracing the actual dynamic process of water (William et al., 2013). The
application of isotope tracers directly relies on isotopic labeling of atmospheric vapor or the resulting
precipitation (Welker et al., 2000; Konstantin et al., 2008). As an effective tool, stable isotope
technology can not only show the relationship between environmental factors and the water cycle
(Araguas-Araguas et al., 1998; Cristhor et al., 2009), water transport and distribution mechanisms (Gao
et al., 2011), and can also deepen the way plants use water (Detjen et al., 2015). And the understanding
of the influence of plant characteristics provides a new observation method for revealing the water
cycle mechanism in the hydrological ecosystem (Nie et al., 2004; Yu et al., 2007; Wang et al., 2019),
and the connection between water use efficiency and water sources (Ehleringe, 1991; Sun et al., 2005;
Chao et al., 2019). Therefore, the application of stable isotope technology provides a new tracer
method for the study of the SPAC hydrological cycle (Zhang et al., 2012; Wen et al., 2017; Pan et al,.
2020). The content of the SPAC hydrological cycle research are dramatically enriches and expands
(Dawson et al., 2002; Ma et al., 2019).

The changes of stable isotopes of soil water are affected by various factors such as atmospheric

precipitation, surface evaporation, soil water migration and vertical movement (Gazi and Feng 2004;
Araguas-Aragua et al., 1995; Jennifer et al., 2015). Also, for desert vegetation, soil water is an essential
water source (Murdock, 1963). Because the isotope ratio in soil moisture obviously changes with depth,
when water is transported between plant roots and stems, it reaches the leaves or young unbolted
branches before its isotopic composition has not changed (Porporato, 2001; Meissne et al., 2014).
Therefore, the content of stable isotopes in soil moisture directly affects the isotopic composition of
water in plants'xylem (Dawson, 1993; Rothfuss et al., 2017). In this way, only by measuring the δD
and δ18O characteristics of plant xylem moisture and soil moisture at different levels can the source of
plant water use be determined (Wu et al., 2015; Meissner et al., 2014; Yang et al., 2014). Precipitation



is an important input factor in the hydrological cycle. The study of the temporal and spatial changes of its isotope characteristics is not only helpful to explore the source of precipitation water vapor and corresponding meteorological and climatic information (Edwards et al., 2010; Daniele et al., 2013; Timsic et al., 2014; Evaristo et al., 2015; Négrel et al., 2016), reflect the historical changes of natural geographic elements (Wei et al., 1994; Speelman et al., 2010; Steinman et al., 2010; Hepp et al., 2015) and climate reconstruction (Thompson et al., 2000; Yao et al., 2008; Xu et al., 2014; Li et al., 2017), but also helpful to determine the hydraulic connection between water bodies (Yao et al., 2009). And combined with the changes of surface water, soil water, and groundwater isotopic composition, can determine the precipitation infiltration and runoff generation process (Bam and Ireso, 2018; Hou et al., 2008), groundwater replenishment, and renewal capacity (Smith et al., 1992; Cortes and Farvolden, 1983), and then lay the foundation for the study of the deep mechanism of the water cycle (Gao et al., 2009). As an important part of the global water cycle, plants control 50-90% of ecosystem evapotranspiration (Jasechko et al., 2013; Coenders-Gerrits et al., 2014; Schlesinger and Jasechko, 2014). Plant roots do not undergo isotopic fractionation when they absorb water (White et al., 1985; Song et al., 2013). Therefore, the water isotope composition of plant roots and stems reflects the isotope composition of water available for plants (Dawson et al., 1991).

We took three different vegetation zones (alpine meadows, forests, and arid foothills) in the Shiyang River Basin as the research object, took the period from April 2018 to October 2019 as the research time, and selected 3 sampling points to analyze the differences and controlling factors of SPAC in different vegetation zones. This research helps to clarify the water use mechanism and local water cycle mechanism of different vegetation zones in tall mountains, and can provide a certain theoretical basis and guiding suggestions for the efficient and reasonable utilization of water resources in arid areas.

**2 Materials and methods**

**2.1 Study area**

The Shiyang River Basin is located at the northern foot of the Qilian Mountains, east of the Hexi Region of Gansu Province (Zhu et al., 2019) (Fig. 1). The Shiyang River originates from the snow-capped mountains on the north side of the Lenglongling in the eastern section of the Qilian



Mountains. The total length of the river is about 250 km, with a basin area of 4.16×104 km². The
annual average runoff is about 15.75×108 km³. The river supply comes from meteoric mountain
precipitation and alpine ice and snow meltwater. The runoff area is about $1.1 \times 10^4$ km², and the drought
index is 1 to 4. The soil is divided into grey-brown desert soil, aeolian sand soil, salinized soil, and
meadow soil. The Shiyang River Basin is located in the hinterland of the mainland. It belongs to a
continental temperate arid climate, with strong solar radiation, sufficient sunshine, short hot summers,
long cold winters, large temperature differences, little precipitation, and strong evaporation. The upper
reaches of the basin is an alpine, semi-arid and semi-humid area, with annual precipitation of 400-600
mm, annual evaporation of 700-1200 mm, and an annual average temperature of 0-4 ℃; the lower
reaches of the basin is a warm and arid area with annual precipitation of 200-400 mm. The annual
evaporation is 1300 ~ 2000 mm, and the annual average temperature is 4 - 8 ℃ (Wen et al., 2013). The
vegetation coverage in the upper and middle alpine areas is relatively good, with trees, shrubs, and
Grassland coverage. The downstream vegetation coverage is poor under the strong influence of
long-term human production and life, mainly desert vegetation.
**2.2 Sample collection**
We have collected samples of precipitation, groundwater, soil, and plant at Lenglong (alpine
meadow), Hulin (forest), and Xiying (arid foothills) in the Shiyang River Basin from April 2018 to
October 2019 (Table 1).
Collection of precipitation samples: Collect precipitation with a rain bucket. The rain measuring
cylinder is composed of a funnel and a storage part. After each precipitation event, the collected liquid
precipitation is immediately devolved to a 100 ml high-density sample bottle. The sample bottle is
sealed with a sealing film, and stored at low temperature. Simultaneously, put a label on the
polyethylene bottle sample, telling the date, types of precipitation (rain, snow, hail, and rainfall). For
the case of multiple precipitation events in one day, multiple sampling is required.
Collection of soil samples: The soil samples are collected with a soil drill at a depth of 100 cm in



the soil at intervals of 10 cm. Put part of the soil sample into a 50 ml glass bottle. The mouth of the
bottle was sealed with parafilm and transported to the observation station for cryopreservation within
10 hours after sampling. It would be used for the determination of stable isotope data. The rest of the
soil sample was placed in a 50 ml aluminum box, and used the drying method to measure the soil
moisture content.
Collection of plant samples: Firstly, collect the xylem stem of the plant with a sampling shear.
Then peel the bark, and put the stem into a 50 ml glass bottle. Lastly, seal the mouth of the bottle and
keep it frozen until the experimental analysis.
Collection of groundwater samples: The groundwater was collected in polyethylene bottles, and
the samples were brought back to the refrigerator at the test station for cryogenic preservation within
10 hours.
**2.3 Sample treatment and analysis**
All the water samples collected are tested with a liquid water analyzer (DLT-100, Los Gatos
Research Center, USA) in the Northwest Normal University laboratory. Each sample and isotope
standard were analyzed by continuous injection six times. To eliminate the memory effect of the
analyzer, we discarded the values of the first two injections and used the average of the last four
injections as the final result value. Isotope measurements are given with the symbol "δ" and are
expressed as a difference of thousandths relative to Vienna Standard Mean Ocean Water:

$$\delta_{(‰)} = \left( \frac{\delta}{\delta - smow} - 1 \right) \times 1000 \qquad (1\text{-}1)$$


Where, $\delta$ is the ratio of $^{18}O/^{16}O$ or $^{2}H/^{1}H$ in the collected sample, $\delta_{v\text{-}smow}$ is the ratio of $^{18}O/^{16}O$
or $^{2}H/^{1}H$ in the Vienna standard sample.
Due to the methanol in plant samples, it is necessary to modify plant samples' original data. Using
different concentrations of pure methanol and ethanol mixed deionized water, combined with Los
Gatos' LWIA-spectral pollutant identification instrument V1.0 spectral analysis software, the
establishment of $\delta^{2}H$ and $\delta^{18}O$ spectral pollutant correction method, determine methanol (NB ) and
ethanol (BB) pollution degree (Meng et al., 2012; Liu et al., 2015). The configuration mode of
methanol and ethanol solution concentration in the correction process is similar to Meng's relevant
experiments (2012). For the broadband metric value NB metric of the methanol calibration result, its



logarithm has a significant quadratic curve relationship with $\Delta\delta^2H$ and $\Delta\delta^{18}O$, and the formulas are
respectively,

$$\Delta\delta^2H = 0.018(\ln NB)^3 + 0.092(\ln NB)^2 + 0.388\ln NB + 0.785(R^2 = 0.991,\ p > 0.0001) \qquad (2\text{-}1)$$

$$\Delta\delta^2O = 0.017(\ln NB)^3 - 0.017(\ln NB)^2 + 0.545\ln NB + 1.356(R^2 = 0.998,\ p < 0.0001) \qquad (2\text{-}2)$$

Its broadband measurements for ethanol correction results in BB metric $\Delta\delta^2H$ and $\Delta\delta^{18}O$ a
quadratic curve and linear relationship, respectively, are:

$$\Delta\delta^2H = -85.67\,BB + 93.664\,(\,R^2 = 0.747\,,\,p = 0.026\,)(\,BB < 1.2\,) \qquad (2\text{-}3)$$

$$\Delta\delta^2O = -21.421\,BB^2 + 39.935\,BB - 19.089\,(\,R^2 = 0.769,\ p < 0.012\,) \qquad (2\text{-}4)$$

**3  Results**
**3.1 The relationship between water stable isotopes in different vegetation zones**
According to the definition of global atmospheric waterline (GMWL) (Craig, 1961), the linear
relationship of $\delta^{18}O$ and $\delta D$ in local precipitation, soil water, plant water, and groundwater is defined as
LMWL, SWL, PWL, and GWL, respectively. By comparing different waterline equations and
analyzing changes in various water bodies, regional meteorological and hydrological conditions can be
determined. The contribution of various environmental factors can be evaluated (Hua et al., 2019).
As shown in Fig. 2, from April 2018 to October 2019, there are certain differences in the
atmospheric waterline equations of different vegetation zones. The slopes of the atmospheric waterline
equations of alpine meadows, forests, and arid foothills are all smaller than that of GMWL. This is
because the study area is located in the northwestern China's arid area, which is weakly affected by the
monsoon, the climate is dry, and the isotopes have undergone strong fractionation. Among them, the
slopes of the atmospheric waterline equations of alpine meadows, forests, and arid foothills are all
smaller than that of GMWL. This is the same as the study area is located in northwestern China's arid
area, which is weakly affected by the monsoon, the climate is dry, and the advection is strong, and the
isotopes have undergone strong fractionation. The slope of the soil waterline in the alpine meadow is
the largest (6.07), and the slope of the soil waterline in the forest (5.10) is greater than the slope of the
soil waterline in the arid foothills (3.94). The intercept has the same characteristics, indicating that the
arid foothills's soil evaporation is the largest. The degree of soil evaporation in alpine meadows is the
smallest. The vegetation coverage area of alpine meadows is larger than that of arid foothills, and the





water retention capacity is better, and soil moisture is not easy to evaporate. The slope of the vegetation
waterline equation in the arid foothills is the largest (2.45), and the slope of the vegetation waterline in
the alpine meadow (1.90) is greater than that of the forest (1.69). The vegetation coverage of the forest
is large, the evaporation is strong, and the evaporation of the vegetation in the arid foothills is relatively
weak.

According to the weighted average of each water body's stable isotope (Table 2), the isotopes of

soil water in alpine meadows are the most depleted and are the closest to the isotopic values of
precipitation. The average isotopic values of groundwater are located between plants and precipitation,
indicating that precipitation is the primary source of replenishment for alpine meadows. The
precipitation isotope of the forest is the most depleted, and the average isotope of groundwater is
between soil water and precipitation but close to precipitation, indicating that forest groundwater is
replenished by soil water and precipitation. The mean isotopic values of soil water in the arid foothills
are between precipitation and groundwater but closer to groundwater, indicating that the soil water in
the arid foothills is mainly supplied by groundwater.
**3.2 Variation of isotope and SWC between different vegetation zone**

The average variation of $\delta^{18}O$ ( $\delta D$ has the same interpretation as $\delta^{18}O$) and SWC in soil water

along the vertical soil profile is shown in Fig. 3. Along the three vegetation zones of alpine
meadow-forest-arid foothills, soil water isotope gradually enriched.The coefficient of variation of the
arid foothills is the largest. The coefficient of variation of the forest is the smallest, indicating that from
forest to arid foothills, it tends to be arider in regions, the greater the coefficient of variation, the greater
the instability of stable isotope soil water. The soil water isotopes of different vegetation zones showed
the same characteristics as the soil depth changed, that is, they were all depleted in May and August,
and enriched in October.

The soil water content of alpine meadows is higher than that of forests and arid foothills, and the

soil water content of alpine meadows increases with the soil depth, while the soil water content of
forests decreases with the soil depth. Forest soil moisture content is caused by the transpiration of the
forest canopy and large water consumption. Compared with forests, plants in alpine meadows have
shallower root systems and smaller canopies, so transpiration and water consumption are lower, and
soil water content is higher. With the continuous progress of vegetation restoration, the vegetation
coverage of alpine meadows will continue to increase, which will reduce soil water evaporation and



increase soil infiltration and water retention capacity. The soil moisture content of alpine meadows and
forests is the largest in August, while the arid foothills's soil moisture content is the smallest in August.
This is because the northern foot of the Qilian Mountains is a windward slope. In August, there is a lot
of rain, and a lot of precipitation falls on the high-altitude alpine meadows and forests. The dry and dry
foothills have little precipitation and low soil water content.
**3.3 Relationship between soil water and plant water in different vegetation zone**
For plants in general, water is absorbed by the root system and moves from root to leaf without
hydrogen and oxygen isotope fractionation (Zhao et al., 2008; Lin et al., 1993). Therefore, by analyzing
the isotopic composition of soil moisture and plant xylem, it is possible to preliminarily determine
whether there is an overlap between soil moisture and plant moisture at different depths (Javaux et al.,
2016; Dawson et al., 2002; Rothfuss et al., 2017; Tetzlaff et al., 2017; McCole et al., 2007; Zhou et al.,
2015; Schwendenmann et al., 2015). Precipitation, surface runoff, and most groundwater are "initial"
sources absorbed by plants after converting into soil water. Before being absorbed by plants, soil water
may undergo evaporation to produce isotopic enrichment, resulting in an increase in the $\delta^{18}O$ and $\delta D$
value of soil water (Chen et al., 2014). Therefore, it can be well explained that the surface soil water in
Fig. 4 is more affluent than the deep soil water.
According to the study area's precipitation, the study area is divided into two time periods: dry
season (October-May of the following year) and the rainy season (June-September) for analysis (Fig. 4).
In the dry season, alpine meadow plants have the most abundant water isotope. There is no overlap
between soil water and plant water. The isotopic values of groundwater and precipitation are similar,
indicating that alpine meadow plants do not directly use soil water in the dry season. In the rainy
season, plant water isotope is the most abundant, and the surface and deep layers of groundwater and
soil water intersect, which indicates that the soil water of the alpine meadow in the rainy season is
mainly recharged by groundwater. In the dry season, due to the low temperature (average temperature
of 0.30°C), there is a large amount of melted ice and snow in the alpine meadow, which is rich in
precipitation and meltwater, and plants do not directly use soil water. In the rainy season (average
temperature 8.72°C), as the temperature increases, plant water isotopes experience intense evaporative
fractionation and are most enriched in isotopes. With the increase of precipitation, runoff, and
formation of groundwater increase, and groundwater supplements soil water. Forest plant water





intersects with deep soil during the dry season and intersects with the soil surface during the rainy
season. This indicates that forest plants mainly use deep soil water during the dry season and shallow
soil water during the rainy season. This is related to the lack of rainfall in the dry season and more
rainfall in the rainy season. During the drought and rainy seasons, the soil water in the arid piedmont
intersects with the groundwater, plant water is enriched, indicating the replenishment relationship
between the soil water and the plant water in the arid piedmont is not apparent. High temperature is
related to groundwater level exposure.
**4 Discussion**
**4.1 The influence of temperature on SPAC**

$\delta^{18}O$ changes significantly with seasons. As shown in Fig. 5, with the changes in the water cycle
of precipitation-soil water-plant water, the $\delta^{18}O$ of forests gradually accumulates, while the soil water
isotopes of arid foothills and alpine meadows are the most depleted in summer. In other seasons, along
precipitation-soil-water-plant water, $\delta^{18}O$ are gradually enriched. In summer, alpine meadows have a lot
of precipitation and large soil water content, but due to low temperature (average temperature in
summer is 9.8°C) and low evaporation, the soil water isotope of alpine meadows is relatively depleted
in summer. In the arid foothills, in summer, especially in August, although the temperature is relatively
high (the average summer temperature is 23.92°C), the soil water content is low, evaporation is weak,
and isotopes are relatively depleted. This phenomenon shows that precipitation plays a major control
role in the water cycle of precipitation-soil-plants. Previous studies have shown that local factors,
especially local temperature mainly control the stable isotope changes of precipitation in mid-latitudes.
If the temperature is below 0°C, the air will expand adiabaticly and the water vapor will change
adiabatic cooling (Rozanski, 1992). When the temperature is between 0°C and 8°C, the influence of
local water vapor circulation is greater. When the temperature is below 8°C, the secondary evaporation

under the clouds is very strong (Ma et al., 2018). Therefore, the temperature is divided into three



gradients (below 0°C, between 0°C and 8°C and above 8°C) to analyze the relationship between
precipitation isotope and temperature. From alpine meadow to arid foothills, the correlations between
temperature and soil are 0.41, 0.30, and 0.19, respectively, and the correlations with plants are 0.24,
0.27, and 0.25, respectively. Compared with precipitation, the temperature effect is not significant. As
shown in Table 3, from the alpine meadow to the arid foothills, the temperature effect of the
precipitation isotope is enhanced, and there is a significant positive correlation with temperature, and
all have passed the significance test. With the increase of temperature, the temperature effect of
precipitation isotope in each vegetation area weakens, and the linear relationship decreases.When the
temperature is lower than 0°C, the correlation between the isotope of precipitation in the arid mountain
foothills and the temperature fails the significance test.When the temperature is between 0°Cand 8°C,
as the temperature increases, the temperature effect of precipitation weakens, which may be related to
the weakening of the local water cycle and the enrichment of precipitation isotopes when the
temperature rises. When the temperature is above 8°C, there is no correlation between the precipitation
isotope and the temperature, but the precipitation isotope value is the most enriched, which may be
related to the isotope enrichment caused by the secondary evaporation under the cloud.
**4.2 The influence of altitude on different vegetation zones**
To analyze the relationship between precipitation isotope and altitude, the study area is divided into
summer half-year (May-September) and winter half-year (October-April of the following year). As the
water vapor mass rises along the hillside, the temperature continues to decrease, and the precipitation
isotope values continue to deplete. As shown in Fig. 6, from the arid foothills to the alpine meadow, the
altitude rises from 1467m to 2097m, and the mean values of precipitation isotopes $\delta^{18}O$ and $\delta D$ have
changed from -7.33‰, -48.62‰ to -9.10‰ and -54.93‰, respectively, and the rate of change was
respectively -7.10‰ , -54.93‰ and   -0.08‰ $(100m)^{-1}$, -0.29‰ $(100m)^{-1}$, this rate of change is within
the globally recognized altitude gradient of precipitation $\delta^{18}O$ is-0.28‰$(100m)^{-1}$ (Poage and



Chamberlain, 2001). In the summer half of the year, the correlation between $\delta^{18}O$ in precipitation and
altitude is -0.97, $R^2$ is 088, and the rate of change is -0.12‰ $(100m)^{-1}$, indicating that there is a
significant negative correlation between $\delta^{18}O$ in precipitation isotope and altitude. And every time the
altitude increases by 100 meters, the $\delta^{18}O$ value of the precipitation isotope changes 0.12‰. In the
winter half of the year, the correlation between $\delta^{18}O$ in precipitation and altitude is -0.95, $R^2$ is 0.79,
and the rate of change is -0.18‰ $(100m)^{-1}$. The correlation between altitude and soil water isotope and
plant water isotope is -0.53 and -0.61, respectively, and their correlation is not as strong as that of
precipitation.
**5 Conclusion**

This paper uses the hydrogen and oxygen isotope method to study the differences and control

factors of SPAC in different vegetation zones. From alpine meadows to forests to arid foothills, as the
altitude decreases, the temperature effect of precipitation isotope increases, and the influence of
temperature increases. When the temperature is lower than 0℃, the temperature effect of the vegetation
zone is the strongest. As the depth increases, soil water isotopes are gradually depleted. The soil water
content of alpine meadows is the largest and increases with the soil depth, while the forest soil water
content decreases with the soil depth, and the soil water content of the arid mountain foothills is the
least in August. In the rainy season, plants mainly use precipitation, while in the dry season, forest
plants mainly use soil water, while alpine meadow plants do not directly use soil water due to the
abundant precipitation and meltwater in the growing season. Exposure of the groundwater level in the
arid mountain foothills can provide water for plants in the dry season. Because forests and grasslands
have the effect of intercepting rainfall, they delay or hinder the formation of surface runoff, and convert
part of the surface runoff into soil flow and groundwater, which can provide part of water resources'
role for plants. To better understand the water cycle of SPAC at different temperatures and altitudes in
high mountain areas, long-term observations of different plants are needed to provide a theoretical
basis for the rational and practical use of water resources in arid mountainous areas.
**Data Availability**

The data that support the findings of this study are openly available in Zhu (2021) at "Data sets of

Stable water isotope monitoring network of different water bodies in Shiyang River Basin, a
typical arid river in China", Mendeley Data, V1, doi: 10.17632/t87pm4b5dx.1



**Author contributation**


Guofeng Zhu and Yuwei Liu conceived the idea of the study; Zhuanxia Zhang analyzed the data;
Zhigang Sun and Leilei Yong were responsible for field sampling;    Liyuan Sang participated in the
experiment; Kailiang Zhao participated in the drawing; Yuwei Liu wrote the paper; Liyuan Sang and
Lei Wang checked and edited language. All authors discussed the results and revised the manuscript.
**Competing interests**
The authors declare no competing interests
**Acknowledgments**
This research was financially supported by the National Natural Science Foundation of China
(41661005, 41867030, 41971036). The authors much thank the colleagues in the Northwest Normal
University for their help in fieldwork, laboratory analysis, data processing.

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

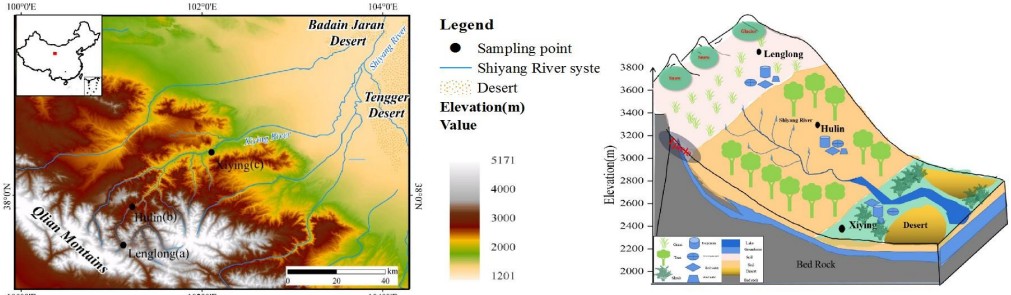


**Figure. 1 Study area and observation system**

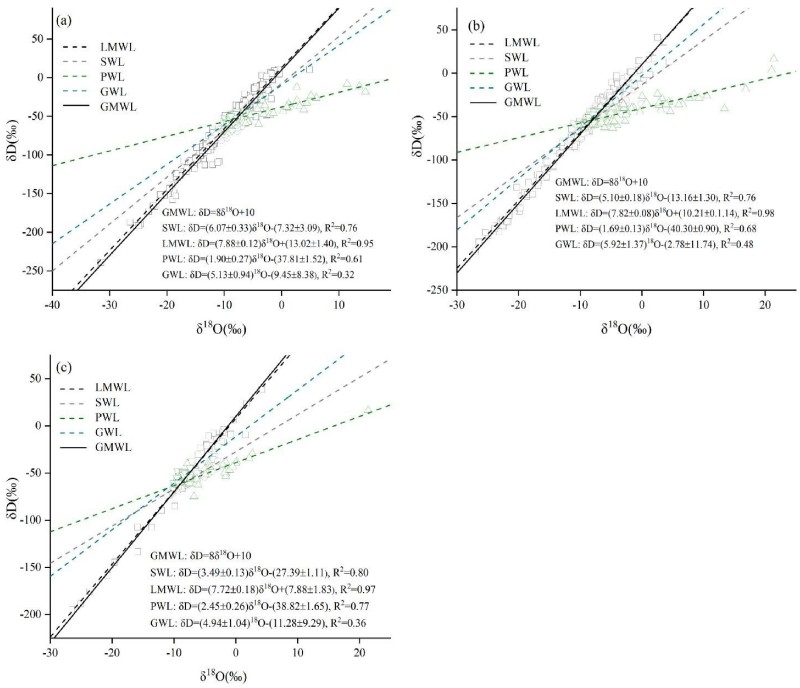


**Fig. 2 Relationship of stable isotopes in different water bodies in alpine meadow (a), forest (b) and arid**

**foothills (c).**


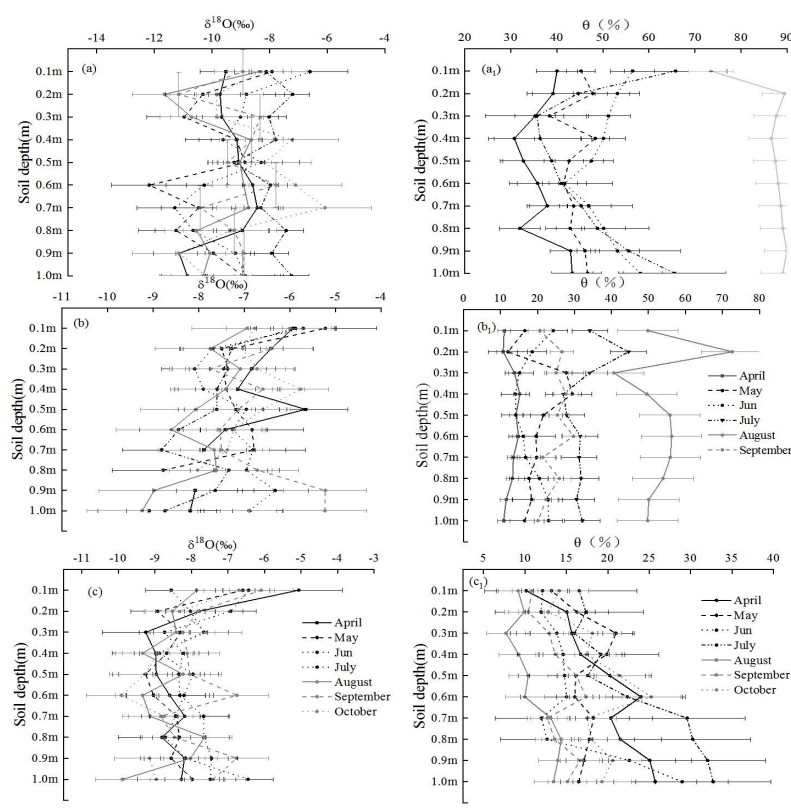


**Fig. 3 The variation of δ¹⁸O and θ (%) with soil depth. (a)-(c) represent alpine meadow, forests and arid**
**foothills, respectively.**

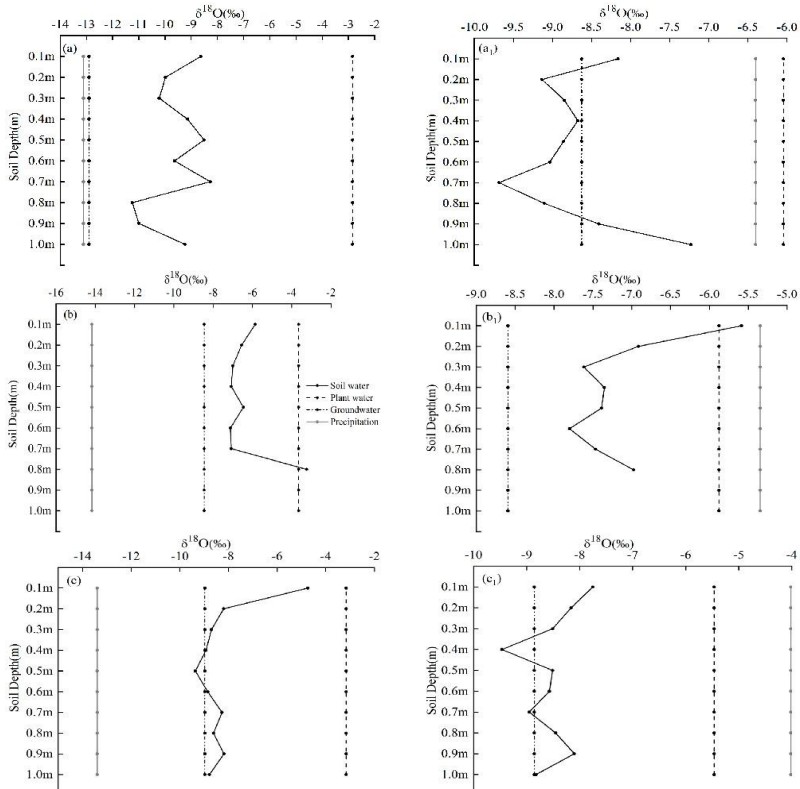

**524**

**525** Fig. 4 (a)-(c) represents the variation of δ¹⁸O of soil, plant, precipitation and groundwater with soil depth in

**526** the alpine meadow, forests and arid foothills in the dry season, and (a₁)-(d₁) represents the variation of δ18O

**527** of soil, plant, precipitation and groundwater in the alpine meadow, forests and arid foothills in the rainy

**528** season.





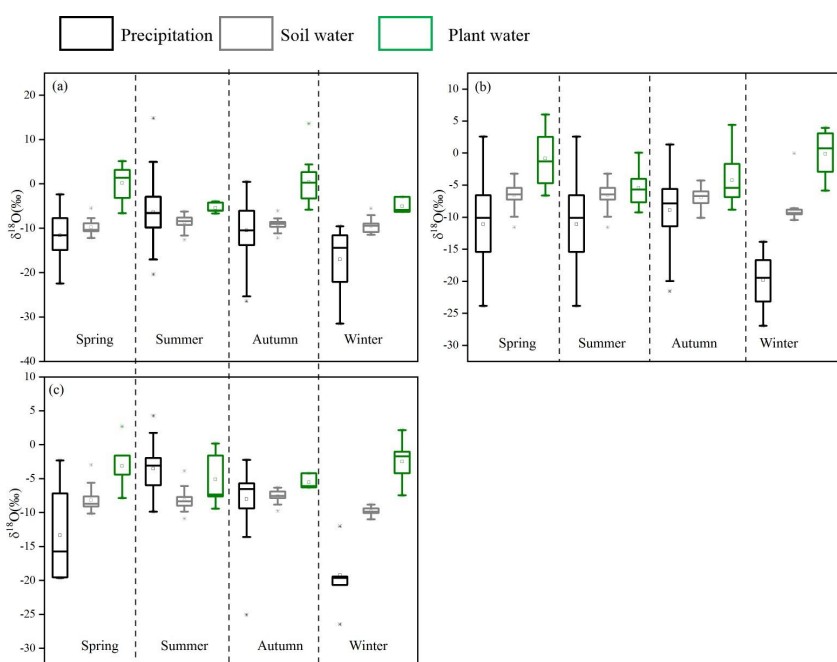


**Fig. 5 Seasonal variations of different water isotopes in alpine meadow (a), forests (b) and arid foothills (c).**

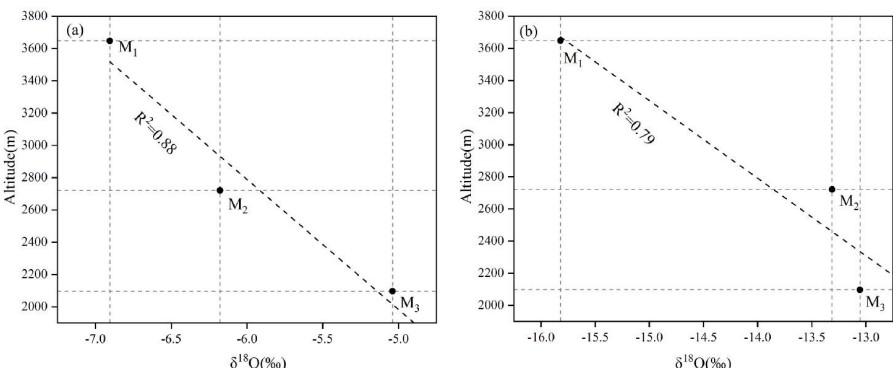


**Fig. 6 Relationship between precipitation isotope and altitude half a year in summer (a ) and half a year in**
**winter (b)**
**Table 1 Basic information table of sampling points**

| Sampling Station | | Geographical Parameters | | | Meteorological Parameters | |
|---|---|---|---|---|---|---|
| | | Longitude (E) | Latitude (N) | Altitude (m) | Temperature (°C) | Precipitation (mm) |
| M1 | Lenglong | 101°50' | 37°33' | 3647 | -0.20 | 1039.10 |
| M2 | Hulin | 101°53' | 37°41' | 2721 | 3.24 | 469.44 |
| M3 | Xiying | 102°18' | 38°29' | 2097 | 7.99 | 194.67 |






| Vegetation zone types | Water types | δ¹⁸O(‰) | | | | δD(‰) | | | |
|---|---|---|---|---|---|---|---|---|---|
| | | Min | Max | Average | Coefficient of Variation | Min | Max | Average | Coefficient of Variation |
| Alpine meadow | Precipitation | -31.49 | 14.79 | -9.44 | -0.70 | -238.62 | 63.43 | -59.43 | -0.84 |
| | Soil water | -12.62 | -5.46 | -9.16 | -0.16 | -83.86 | -26.13 | -62.92 | -0.16 |
| | Plant water | -6.68 | 5.12 | -1.68 | -2.18 | -60.22 | -12.14 | -41.14 | -0.28 |
| | Groundwater | -10.07 | -7.71 | -8.84 | -0.07 | -68.55 | 43.72 | -54.85 | -0.10 |
| Forest | Precipitation | -26.96 | 4.38 | -8.63 | -0.74 | -205.40 | 41.35 | -60.24 | -0.87 |
| | Soil water | -11.96 | -0.07 | -7.01 | -0.25 | -78.43 | -18.48 | -48.68 | -0.21 |
| | Plant water | -9.24 | 5.98 | -5.44 | -1.31 | -63.29 | -23.77 | -45.12 | -0.24 |
| | Groundwater | -10.25 | -7.43 | -8.56 | -0.09 | -68.80 | -43.75 | -53.46 | -0.12 |
| Arid foothills | Precipitation | -26.47 | 4.24 | -7.50 | -0.87 | -194.34 | 38.62 | -48.62 | -1.04 |
| | Soil water | -10.98 | -2.96 | -8.23 | -0.15 | -74.22 | -8.79 | -59.17 | -0.12 |
| | Plant water | -9.41 | 2.67 | -3.61 | -0.88 | -74.90 | -29.39 | -48.79 | -0.23 |
| | Groundwater | -10.34 | -7.43 | -8.88 | -0.07 | -71.67 | -44.26 | -55.12 | -0.09 |

**Table 2 Comparison of stable isotope of water in different vegetation zones**


**Table 3 Correlation between precipitation isotopes and different temperatures in different vegetation zones**

| Vegetation zone type | Correlation below 0°C (δ¹⁸O /δD) | Correlation between 0°C-8°C (δ¹⁸O /δD) | Correlation above 8°C (δ¹⁸O /δD) | Correlation during the study period |
|---|---|---|---|---|
| Alpine meadow | 0.51*/0.59* | 0.30*/0.24* | 0.15/0.12 | 0.59*/0.61* |
| Forest | 0.95*/0.94* | 0.66*/0.69* | 0.14/0.10 | 0.69*/0.65* |
| Arid foothills | 0.47/0.51 | 0.79*/0.71* | 0.31/0.14 | 0.83*/0.81* |
