# Peer review of "Isotopic differences of soil-plant-atmosphere continuum composition and control factors of different vegetation zones"

_Biogeosciences, 2021_

## Author Comment (AC1)

Dear Reviewer:

Thank you for your letter and for the reviewer's comments concerning our manuscript entitled "Difference of SPAC composition and control factors of different vegetation zones in north slope of Qilian Mountains" (Manuscript Number: bg-2021-127).

According to the comments of the reviewer, we have revised our manuscript carefully. The primary corrections and the response to the reviewers' comments are as follows.

**Responses to the reviewer's comments:**

**This manuscript (bg-2021-127), entitled "Difference of SPAC composition and control factors of different vegetation zones in north slope of Qilian Mountains", investigated the differences in SPAC composition of three different vegetation zones of the period of April 2018 to October 2019, and attempled to find the controlling factors. This paper is the one that I have seen so far in the research on SPAC water cycle with relatively rich data and very reasonable argumentation. This manuscript provides a lot of valuable isotope data for understanding the water cycle in arid and semi-arid regions. In this paper, the micro-SPAC water cycle is well correlated with the basin water cycle, and the case studies are compared and discussed at the regional and global scales. It is a writing style that I admire a great deal. So I support the publication of the article. However, there exist some problems, including English language, basic format, and the description of the research samples is not enough. In addition, many explanations in the text are not clear enough. A good article must be very scientific and well expressed. Therefore, I think the article can only be published after having solve the following problems.**

Response: We have gradually improved it by carefully revising every recommendation you mentioned.

**Specific comments:**

1. **Why choose these three sampling points as the research area of this manuscript? How long is the sampling interval? The author has made a lot of**

**descriptions on sample collection and experimental analysis, but if the author can more clearly describe how to select samples, it will be more reasonable, for example, in Section 2.2: We have collected samples of precipitation, groundwater, soil, and plant at Lenglong (alpine meadow), Hulin (forest), and Xiying (arid foothills) in the Shiyang River Basin from April 2018 to October 2019.**

Response: We consider the sea surface, topography, the distance between sampling points, and the consistency of sampling time. We finally chose the Lenglong sampling point to represent the alpine meadow, the Hulin sampling point represents the forest, and the Xiying sampling point represents the arid foothills. Our sample collection interval is one month. From April 2018 to October 2019, we collected a total of 1281 samples in the Shiyang River Basin, including 472 precipitation samples, 570 soil samples, 119 plant samples, and 120 groundwater samples. Soil and plant samples need to be vacuum extracted. The extraction instrument used is the LI-2100 automatic vacuum condensation extraction system. The analysis instrument is the LWIA-24d liquid water isotope analyzer. For precipitation and groundwater samples, they are directly analyzed with a fluid water isotope analyzer. After the test of each batch of samples is completed, we will use the LIMA software to select the wrong pieces and re-analyze the experiment until the tested value passes the LIMA software inspection.

2. **Section 3.2: With the continuous progress of vegetation restoration, the vegetation coverage of alpine meadows will continue to increase. How do you know that the vegetation coverage of alpine meadows has increased? Is the conclusion drawn from the relevant literature or based on actual data?**

Response: The increase of vegetation coverage rate of the alpine meadow is calculated according to the rise in grassland area, with data from the comprehensive natural survey report of Shiyang River Basin. According to the complete survey report on the natural resources of the Shiyang River Basin, the natural grassland area of the Shiyang River Basin was 12,452.72 hectares in 2017 and 13,071.94 hectares in 2019. We can see that from 2017 to 2019, the grassland area of the Shiyang River Basin increased by 619.22 hectares, which shows the increase in vegetation coverage in the

Shiyang River Basin.

3. **Section 3.2: "Along the three vegetation zones of alpine meadow-forest-arid foothills, soil water isotope gradually enriched.The coefficient of variation of the arid foothills is the largest". Why is the coefficient of variation of the isotope of soil water in the dry mountain foothills the largest?**

Response: On the one hand, the stable isotope variation coefficients of different water bodies in the three vegetation areas calculated through mathematical statistics are shown in Table 2. In Table 2, the coefficient of variation of soil water isotope of the alpine meadow is -0.16, the coefficient of variation of soil water isotope of the forest is -0.25, and the coefficient of variation of soil water isotope of the arid foothills is -0.15. It can be known from the numerical value that from forest to arid foothills, the more inclined to arid regions, the larger the coefficient of variation of soil water isotope, and the more unstable the soil water isotope. On the other hand, there is little precipitation in the arid foothills, and the soil water is mainly recharged by groundwater. Precipitation events mainly occur in summer when the temperature is higher. The isotope distillation of soil water is significant, and the isotope of soil water is very different when there is precipitation and when there is no precipitation so that the isotope of soil water in the arid foothills varies greatly, and the coefficient of variation is large.

4. **Section 3.2: "The soil water content of alpine meadows is higher than that of forests and arid foothills, and the soil water content of alpine meadows increases with the soil depth". Why does the soil water content of alpine meadows increase with the increase of soil depth? The author did not give an explanation in the manuscript. Does this characteristic of change appear in different soil depths?**

Response: There is a lot of precipitation in the alpine meadow area, even in the winter, when the temperature is below zero, there is solid precipitation in Lenglong. Alpine meadow vegetation has a shallow root system, small tree crown, low transpiration and low water consumption. The low temperature throughout the year and high air humidity make it difficult for the soil water in alpine meadows to evaporate. As the

soil depth increases, there is a frozen soil layer. The lower temperature makes the firm soil layer almost always exist, making the soil water content of the alpine meadow increase with the soil depth. The characteristics of this change are different in different soil layers of alpine meadows. The following describes the changes of soil water content in different soil layers of alpine meadows:

In April, the soil water content of alpine meadows decreased at 10-40 cm, gradually increased at 40-70 cm, and progressively reduced at 80-100 cm. In May, the soil water content of the alpine meadow increases at 10-20 cm, decreases at 20-30 cm, increases at 30-40 cm, decreases at 40-60 cm, and gradually increases at 60-100 cm. In June and July, the soil water content of alpine meadows decreased by 10-60 cm and increased by 70-100 cm. In August, the soil water content of the alpine meadow increased by 10-40cm, decreased by 40-60cm, and increased by 60-100cm. In September, the soil water content of alpine meadows decreased by 10-30cm, increased by 30-50cm, and decreased by 60-100cm. So the soil water content of alpine meadows at the same depth varies significantly in different months, and the changes in soil water content of alpine meadows to varying depths at the same time are more complicated. But in general, the soil moisture content of alpine meadows increases with the increase of soil depth, and the forest soil moisture content decreases with the growth of soil depth.

5. **Section 3.3, line 228-229: "High temperature is related to groundwater level exposure". What does this sentence mean?**

Response: The original meaning of this sentence is that the soil water and groundwater in the arid mountain foothills intersect, which is related to the exposure of the groundwater level caused by high temperature. We have rewritten the relevant part:

In Fig. 4, in the rainy season, the surface layer of soil water in the arid foothills intersects with plant water, and the surface and deep layers of groundwater intersect with soil water, and precipitation is the most abundant. It shows that the plant water in the dry mountain foothills in the rainy season preferentially uses the surface water of the soil and does not directly use the precipitation. The soil water

mainly supplies the groundwater. In the dry season, plant water is most abundant, and the isotopic values of groundwater and soil water are close. It shows that the soil water in the dry mountain foothills is mainly recharged by groundwater in the dry season. According to the natural resources survey report of the Shiyang River Basin, the buried groundwater level in the arid piedmont area is 2.5-15 meters, and the groundwater burial is relatively shallow, making the soil water in the arid foothills mainly recharged by groundwater in the dry season.

6. **In the discussion section, why only discuss the impact of temperature and altitude on SPAC? Why are other factors not discussed?**

Response: Dai et al. (2020) pointed out that local factors, especially local temperature, mainly control the change of stable isotope precipitation in mid-latitudes. In this study, the Shiyang River Basin is located in the arid area of the northwest inland. The isotope precipitation effect is not significant, and the temperature effect is substantial. And because the Shiyang River Basin is mainly affected by advection water vapor, the elevation effect of isotopes is obvious. Therefore, only the influence of temperature and altitude on SPAC will be discussed in the Discussion. In order to compare temperature and altitude with other factors, we added the influence of other control factors on the isotope composition of SPAC in the discussion. To compare temperature and altitude with other factors, we added the influence of other control factors on the isotope composition of SPAC in the Discussion.

7. **Line 240-241: This phenomenon shows that precipitation plays a major control role in the water cycle of precipitation-soil-plants. What does this phenomenon mean?**

Response: This phenomenon means that along with the change of precipitation-soil water-plant water, forest $\delta^{18}O$ is gradually enriched. In contrast, soil water isotopes in arid foothills and alpine meadows are most depleted in summer. In other seasons, $\delta^{18}O$ is enriched along precipitation-soil water-plant water. In summer, the alpine meadow has a lot of precipitation and the soil has a lot of water. However, due to the low temperature (average summer temperature of 9.80℃), water is not easy to evaporate, and soil water isotopes of alpine meadows are relatively depleted in summer. In

summer in the arid foothills, especially in August, although the temperature is relatively high (the average summer temperature is 23.92℃), the precipitation is low, the soil water content is low, and the soil water isotope is somewhat depleted. So soil water isotope and plant water isotope are affected by precipitation isotope, and precipitation isotope plays a significant control role in precipitation-soil water-plant water.

8. **Line 241-242: "Previous studies have shown that local factors, especially local temperature mainly control the stable isotope changes of precipitation in mid-latitudes". The author does not mention any information about the previous research in the manuscript, such as the source of the research.**

Response: We have added the publication information of the reference.

Hans, P. P. 2009. Data Transformation in Statistical Analysis of Field Trials with Changing Treatment Variance. *Agronomy Journal*(4), doi:10.2134/agronj2008.0226x.

9. **Line 248-250: "From alpine meadow to arid foothills, the correlations between temperature and soil are 0.41, 0.30, and 0.19, respectively, and the correlations with plants are 0.24, 0.27, and 0.25, respectively". How are these correlation coefficients obtained?**

Response: These correlation coefficients are obtained by analyzing the correlation between temperature and soil and plants in alpine meadows, forests, and arid foothills. Because the temperature effects of soil water isotope and plant water isotope are not as significant as precipitation isotope, and precipitation isotope plays an important control role in SPAC, the description of soil water isotope and plant water isotope temperature effect is less important in this article.

10. **Section 4.2: In this part, the author mainly discusses altitude and precipitation isotope, but the description of the relationship between soil water isotope and plant water isotope and altitude is too little.**

Response: We have added a description of the relationship between plant water isotope and altitude and the relationship between soil water isotope and altitude.

The study area is divided into the rainy season (May-September) and dry season

(10-April of the following year), and the relationship between altitude and isotope is analyzed (Fig. 7). The altitude effect of precipitation isotope is stronger than the relationship between soil water isotope and altitude and the relationship between plant water isotope and altitude, but the relationship between plant water $\delta D$ and altitude in the rainy season is stronger than the relationship between soil water $\delta D$ and altitude. It shows that in SPAC, precipitation isotope is most affected by altitude, and plant water isotope is least affected by altitude. As the quality of water vapor rises along the hillside, the temperature continues to decrease, and the isotopic values of precipitation continue to be consumed. From the arid foothills to alpine meadows, the elevation rises from 2097m to 3647m. The average values of precipitation isotopes $\delta^{18}O$ and $\delta D$ changed from -7.33‰ to -9.10‰, and from -48.62‰ to -54.93‰, respectively. The rate of change was -0.11‰(100m)$^{-1}$, -0.41‰(100m)$^{-1}$ , In the globally recognized precipitation $\delta 18O$ altitude gradient range, this rate of change is -0.28‰ (100m)$^{-1}$ (Porch and Chamberlain, 2001). The squares of correlation coefficients between $\delta^{18}O$ and $\delta D$ of rainy season precipitation and altitude are 0.79 and 0.98. The rate of change is -0.12‰(100m)$^{-1}$ and -1.05‰(100m)$^{-1}$, respectively. In the dry season, the correlation coefficient squares of $\delta^{18}O$ and $\delta D$ with altitude are 0.88 and 0.90, respectively, and the rate of change is -0.18‰(100m)$^{-1}$ and -0.79‰(100m)$^{-1}$, respectively. It can be seen that the altitude effect of precipitation $\delta^{18}O$ is stronger in the dry season ($R^2$=0.88) than in the rainy season ($R^2$=0.79), and the altitude effect of precipitation $\delta D$ is stronger in the rainy season ($R^2$=0.98) than in the dry season ($R^2$=0.90). The relationship between soil water isotope and altitude is stronger in the

rainy season ($R^2$=0.26, $R^2$=0.73) than in the dry season ($R^2$=0.28, $R^2$=0.26). The relationship between plant water $\delta^{18}O$ and altitude is stronger in the dry season ($R^2$=0.11) than in the rainy season ($R^2$=0.11), and the relationship between plant water $\delta D$ and altitude in the rainy season ($R^2$=0.62) is stronger than that in the dry season ($R^2$=0.56). It can also be seen from the figure that there are anti-elevation shows in some areas, mainly from forests to dry foothills. This may be related to the existence of reservoirs in the arid foothills. Reservoirs may cause the reversal of the local water vapor cycle-the anti-elevation effect. Generally speaking, there is a negative correlation between altitude and SPAC isotope composition. The altitude effect of precipitation isotope is stronger than the relationship between soil water isotope and altitude, and stronger than the relationship between plant water isotope and altitude.

[Figure]

11. **Line 255-256: "...the correlation between the isotope of precipitation in the arid mountain foothills and the temperature fails the significance test." Why fails the significance test?**

Response: When the temperature is lower than 0 degrees celsius, there may be solid precipitation in the alpine meadows in the upper reaches of the Shiyang River Basin. At the same time, there is almost no precipitation in the middle and lower reaches. As the precipitation samples in the arid foothills are few and mainly concentrated in summer, when the temperature is lower than 0 degrees celsius, the correlation between precipitation isotope and temperature failed the significance test. In future research, it is necessary to strengthen the long-term systematic observation of the middle and lower reaches of the Shiyang River Basin, especially the arid piedmont area, and obtain more precipitation samples below 0°C, to more accurately analyze the impact on precipitation isotope.

**Minor comments:**

**1. In the abstract, the author directly proposed SPAC, the author did not give any explanation about wahst is SPAC.**

Response: We have added a description of SPAC in the abstract.

**2. Line 28: the reference is incorrectly cited.**

Response: The order of reference has been modified.

Song et al. 2002; Gao et al., 2009; Coplen, 2013; Shou et al. 2013

**3. Part 2.3, the font size of the formula (1-1) , (2-1) to (2-4) are different.**

Response: We have unified the font and size of all formulas in this article.

**4. Line 47: change "The content of the SPAC hydrological cycle research are dramatically enriches and expands" to "The content of SPAC hydrological cycle research has been greatly enriched and expanded".**

Response: We have adjusted this sentence according to your suggestion.

**5. In the manuscript, some formats are not standardized, and some letters**

**should be superscripts, such as in line 34, 18O ; line 57, δ18O; line 88, 15.75×108.**

Response: We have standardized all the formats in the text.

**6. The scientific counting method of temperature in this manuscript is inconsistent, as in section 4.1: 9.8℃, 23.92℃.**

Response: We have unified the scientific counting method of all the numbers in the article.

**7. Line 88: change 15.75×108 km3 to 1.58×10⁸ km³.**

Response:  We have changed 15.75×10⁸ km3 to 1.58×10⁸ km3.

**8. Lin 89-91: please add the corresponding references.**

Response: We have added the publication information of the reference.

Zhou, J. J.,Zhao, Y. R., Huang, P., and Liu. C. F. 2020. Impacts of ecological restoration projects on the ecosystem carbon storage of inland river basin in arid area, China. Ecological Indicators. doi:10.1016/j.ecolind.2020.106803.

**9. Improve the clarity of Figure 1 to Figure 4 to make it easier to obtain relevant information from the diagram.**

Response: We have adjusted the clarity of the pictures, and re-drawn some pictures to facilitate readers to obtain relevant information from the pictures more intuitively.The revised figure is shown below:

[Figure]

Fig. 1 Study area and observation system

[Figure]

Fig.3 Relationship of stable isotopes in different water bodies in alpine meadow (a), forest (b) and arid foothills (c)

[Figure]

Fig. 4 (a)-(c) represents the variation of $\delta^{18}O$ of soil, plant, precipitation and groundwater with soil

depth in the alpine meadow, forests and arid foothills in the dry season, and (a₁)-(d₁) represents the variation of δ18O of soil, plant, precipitation and groundwater in the alpine meadow, forests and arid foothills in the rainy season

[Figure]

Fig.5 The variation of $\delta^{18}O$ and soil water content ($\theta$, %) with soil depth. (a)-(c) represent alpine meadow, forests and arid foothills, respectively

[Figure]

Fig. 6 Seasonal variations of different water isotopes in alpine meadow (a), forests (b) and arid foothills (c)

[Figure]

Fig. 7 Relationship between different isotope and altitude in the dry season (a, c ) and in the rain

season (b, d), M₁ stands for alpine meadows, M₂ stands for forests, and M₃ stands for arid foothills

**10. Line 226 and line 228: In order to maintain consistency, please change "arid piedmont" to "arid foothills".**

Response: We have unified the terminology in the text.

**11. Line 256: Please pay attention to the spaces between the text , such as change "0°Cand" to "0°C and'.**

Response: We have unified all the spaces in the text.

---

## Author Comment (AC2)

Dear Reviewer:

Thank you for your letter and for the reviewer's comments concerning our manuscript entitled "Difference of SPAC composition and control factors of different vegetation zones in north slope of Qilian Mountains" (Manuscript Number: bg-2021-127).

According to the comments of the reviewer, we have revised our manuscript carefully. The primary corrections and the response to the reviewers' comments are as follows.

**Responses to the reviewer's comments:**

**In this manuscript, Liu et al. explored oxygen and hydrogen isotope signals in precipitation, groundwater, soil water and plant water in different vegetation zones within the upper reach of Shiyang River Basin, and aimed to elucidate internal linkages between various water bodies. Such investigation could deepen our understanding on mechanisms in water cycle and facilitate ecosystem management of water-limited areas. The topic of manuscript falls into the scientific scope of Biogeosciences. The authors selected three representative vegetation zones in the study area, and worked hard to collect many samples last for more than one year. I believe their data are informative and interesting. However, in my view, the manuscript is not well written, including the whole structure, data analysis, interpretation of results, discussion, and language issues. Thus, the current version of this manuscript is beyond the standard of BG, and I suggest the authors resubmit the manuscript after major revision.**

Response: We have gradually improved it by carefully revising every recommendation you mentioned.

**Major concerns**

**1. It is difficult to figure out the background and necessity of this study from the current version. In the initial paragraph of Introduction, the authors only told us that water isotopes were useful with so many sentences, but they didn't show us the latest developments and trends of isotopic research in soil-plant-atmosphere continuum. In the second paragraph, the authors described soil water,**

**precipitation, and plant water. But they didn't raise any scientific problems. Until the last paragraph, the authors still didn't state the reasons to perform research in Shiyang River Basin.**

Response: Thank you very much for your suggestion. In the second paragraph of the Introduction, we added the latest developments and trends in isotopic research on the soil-plant-atmosphere continuum:

The research of the water cycle based on SPAC plays a vital role in the study of water in arid areas and the sources of plant water use (Price et al., 2012; Shou et al., 2013). Hydrogen and oxygen stable isotope methods have been used to study the water cycle at the interface of "soil-root", "soil-plant" , and "soil-atmosphere", but only a small number of parameters play an important role in the complex interactions of various surfaces (Durand et al., 2007; Deng et al., 2013; Li et al., 2006; West et al., 2006 ). At present, the study of stable hydrogen and oxygen isotopes is no longer limited to a single aspect of the SPAC interface water cycle (Zhang et al., 2016; Penna et al., 2020). The tracer study of oxygen isotopes in soil water-plant water-plant fossils in steppe has been carried out internationally, providing a theoretical basis for studying the spatial distribution of oxygen isotopes in soil water and palaeoclimate (Webb et al., 2003). However, the study of the SPAC water cycle as a whole has not been carried out. In future research, the application of hydrogen and oxygen stable isotope technology to the whole system of "five water conversion" of precipitation, surface water, groundwater, soil water and plant water is a new field worth exploring (Gao et al., 2017; Tipple et al., 2017), which will ultimately solve some core problems in the process of the water cycle and production practice problems (Gao et al., 2017; Peng et al., 2010; Tipple et al., 2017). Through research in different water bodies, such as the composition of hydrogen and oxygen isotope, can further understanding the mechanism of vegetation using water in different water bodies of water (Huang et al., 2012; Yang et al., 2015), such as the migration and transformation of relations between to solve ecological water requirement for vegetation construction in arid and semiarid areas and some key scientific problems of vegetation restoration and provide a scientific basis for ecological environment construction in western (Wu et al., 2016;

2017). In the existing research, how to extend the small-scale SPAC water cycle research results to the large-scale area has become a hot spot and difficulty in the current research.

We rewrote the third part of the Introduction and added the scientific questions of this article in this part:

The Shiyang River Basin is the river basin with the greatest ecological pressure and the most severe water shortage in China. Due to the lack of water resources and the small exchange of energy and water with the outside world, the hydrological cycle is mainly based on the vertical circulation of groundwater-soil water-atmospheric water. The purpose of this study is to: (1) analyze the SPAC water cycle process in different vegetation areas; (2) determine the potential factors that control the SPAC water cycle. The research is helping to clarify the water resource utilization mechanism and the local water cycle mechanism of different vegetation areas in high mountainous areas and provide a certain theoretical basis and guiding suggestions for the practical and reasonable use of water resources in arid areas.

**2. In the current version, Introduction, Results, and Discussion sections were mismatched from each other. From Introduction, I understand the authors intend to analyze the isotopic differences in different bodies and the potential controlling factors. However, no data on the controlling factors were shown in Results, while temperature and altitude were discussed in Discussion, which were not mentioned before. I suggest to reorganize the manuscript and closely link each section.**

Response: We have adjusted the manuscript's structure so that the introduction, results, and discussion parts match each other. In the results, we modify part 3.1 to Changes in meteorological parameters over time; modify part 3.2 to The relationship between stable water isotopes in different vegetation zones; modify part 3.3 to Relationship between soil water and plant water in different vegetation zones; modify part 4.1 to Variation of soil isotope and s between different vegetation zones; modify part 4.2 to Control factors of SPAC in different vegetation zones. In the Result, we added information on control factors:

Soil samples were placed in a 50 ml aluminum box, and the drying method determined soil moisture content. Meteorological data, including precipitation, relative humidity and temperature, are obtained from a meteorological station in the Shiyang River Basin. Figure 2 shows the changes in daily precipitation, relative humidity, temperature and soil water content (SWC) in the study area from April 2018 to October 2019. During the summer monsoon (April to September), the accumulated precipitation accounted for 90.4% of the total precipitation, and the average daily precipitation on rainy days was 3.98 mm. During the winter monsoon (October to March), the accumulated precipitation accounted for 9.6% of the total precipitation, and the average daily precipitation on rainy days was 0.13 mm. During the observation period, the temperature from -16.2℃ to 32℃, and the average temperature of summer monsoon and winter monsoon were 20.20℃ and -0.69℃, respectively. The average SWC value of 0-100cm soil layer varies from 2.58% to 89.96 %, and the low SWC value usually appears in summer, which is related to the strong evaporation of soil and the strong transpiration of vegetation.

[Figure]

Fig. 2 Diurnal variation of relative humidity, precipitation, temperature, and

**3. A subsection on data analysis is also necessary in the Materials and Methods section. In Results, I found many words like "…smaller than…", "…greater than…", "…the smallest…", and "…closer to…". Such presentations need solid evidence of statistical analysis. So do the comparisons of slopes and intercepts between different water lines. It will also be better to show the quantitative relationships between isotopic information of SPAC and potential controlling factors.**

Response: We have added a data analysis section in Materials and Methods.

Since the isotopic data are generally distributed according to the Kolmogorov-Smirnov (KS) test, Pearson correlation is performed to describe the various correlations between different water types (for example, precipitation, soil water, plant water, and groundwater) and the relationship between isotopes in different vegetation zones and control factors. The significance level for all statistical tests was set to the 95% confidence interval. All statistical analyses were performed using SPSS software.

In results, we added relevant statistical analysis data to support the mentioned words like … smaller than…", "…greater than…", "…the smallest…", and "…closer to… and the comparisons of slopes and intercepts between different water lines. We also added relevant content about the quantitative relationship between isotopic information of SPAC and potential controlling factors.

**Specific comments**

**1. L1: As first mentioned in the title, the abbreviation SPAC should be written as its complete form, soil-plant-atmosphere continuum. Also, the authors may want to say "Isotopic differences of…".**

Response: We have added the full form of SPAC based on your suggestion and revised the title. The new title is as follows:

Isotopic differences of soil-plant-atmosphere continuum composition and control factors of different vegetation zones in north slope of Qilian Mountains

**2. L15: "The results showed…".**

Response: We have modified it according to your suggestion.

**3. L26: "…changes of oxygen and hydrogen isotopes in water…".**

Response: We have modified it according to your suggestion.The revised sentence is as follows:

The relative abundance changes of oxygen and hydrogen isotopes in water technology in water can indicate the water cycle and water use mechanism in plants, so isotope technology has become an increasingly important method for studying the water cycle (Gao et al. 2009; Song et al. 2002; Coplen, 2013; Shou et al. 2013)

**4. L41: "…, but also…".**

Response: We have modified it according to your suggestion. The revised sentence is as follows:

As an effective tool, stable isotope technology can not only show the relationship between environmental factors and the water cycle (Araguas-Araguas et al., 1998; Cristhor et al., 2009), water transport and distribution mechanisms (Gao et al., 2011), and but also deepen the way plants use water (Detjen et al., 2015).

**5. L43: Nie et al., 2014.**

Response: We have modified it according to your suggestion. The revised sentence is as follows:

And the understanding of the influence of plant characteristics provides a new observation method for revealing the water cycle mechanism in the hydrological ecosystem (Nie et al., 2014; Yu et al., 2007; Wang et al., 2019) and the connection between water use efficiency and water sources (Ehleringe, 1991; Sun et al., 2005; Chao et al., 2019).

6. **L46: It seems that the "SPAC" appears suddenly here. Necessary conversions of words and content are needed.**

Response: We have added the latest developments and trends in isotopic research on the soil-plant-atmosphere continuum before this sentence as a conversion of the following sentence.

**7. L49: Both "soil water" and "soil moisture" are used in the manuscript. I suggest use one of them.**

Response: We have unified the term as soil water in the manuscript.

**8. L51: Is sampling of the current study involving desert vegetation?**

Response: The sampling in the current study does not involve desert vegetation. The purpose of adding this sentence is to show that the closer to arid areas, the more important soil water is to plant water. We have deleted this sentence .

**9. L56-58: "The source of plant water use can be determined by measuring…".**

Response: We have modified this sentence according to your suggestion. The revised sentence is as follows:

The source of plant water use can be determined by measuring the δD and $\delta^{18}O$ characteristics of plant xylem moisture and soil moisture at different levels (Wu et al., 2015; Meissner et al., 2014; Yang et al., 2014).

**10. L66-70: The sentence is too long, and specific subjects before "can" and "lay" lack.**

Response: We have reorganized this sentence. The revised sentence is as follows:

Combined with changes in the isotopic composition of surface water, soil water and groundwater, the process of precipitation infiltration and runoff generation can be determined (Bam and Ireso, 2018; Hou et al., 2008) and groundwater recharge and regeneration capacity (Smith et al., 1992; Cortes and Farvolden, 1983). Furthermore, it lays a foundation for studying the deep mechanism of the water cycle (Gao et al., 2009).

**11. L70: "plant transpiration" or "vegetation transpiration".**

Response: We have revised this paragraph based on your suggestion. The revised sentence is as follows:

As an important part of the global water cycle, plants control 50-90% of plant evapotranspiration (Jasechko et al., 2013; Coenders-Gerrits et al., 2014; Schlesinger and Jasechko, 2014)

**12. L90: What kind of the drought index?**

Response: The drought index here is the ratio of annual evaporation capacity to annual precipitation. The formula is: $r = E_0 / P$, which is an index reflecting the degree of climatic drought. $E_0$ is the annual evaporation capacity, the unit is mm. It is often

replaced by E-601 water surface evaporation. In the comprehensive zoning of my country's drought index: r between 0.5 and 1 is a humid zone; r between 1 and 3 is a semi-humid zone ; r between 3 and 7 is a semi-arid zone and r is greater than 7 is an arid zone.

**13. L90: "…classified as…", not "divided into".**

Response: We have modified this sentence. The revised sentence is as follows:

The soil is classified as grey-brown desert soil, aeolian sand soil, salinized soil, and meadow soil.

**14. L91-93: Replace these descriptions with exact data.**

Response: We have replaced these descriptions with exact data The revised sentence is as follows:

The Shiyang River Basin is located in the hinterland of the mainland. It has a continental temperate arid climate with strong solar radiation. The annual average sunshine hours are 2604.8-3081.8 hours, the annual average temperature is -8.20-10.50℃, the temperature difference between day and night is 25.2℃, the annual average precipitation is 222 mm, and the annual average evaporation is 700-2000 mm.

**15. L96-97: Add to the previous sentence.**

Response: We have added this paragraph to the previous sentence. The revised sentence is as follows:

the lower reaches of the basin is a warm and arid area with annual precipitation of 200-400 mm, annual evaporation is 1300 - 2000 mm, and the annual average temperature is 4 - 8 ℃ (Wen et al., 2013).

**16. L98: Did the authors investigate vegetation in the study area? If yes, please show the data of vegetation coverage; if not, relevant references should be added. In addition, "relatively good" is not a proper expression in scientific papers.**

Response: Our research team has investigated the vegetation in the study area. We have added reference materials published by the research team and revised this sentence based on your suggestions. The revised sentence is as follows:

The vegetation coverage in the upper and middle alpine regions is better, with trees, shrubs, and grass covered (Wan et al., 2019).

Wan, Q. Z.,  Zhu, G. F., Guo, H. W., Zhang, Y., Pan, H. X ., and Yong, L. L et al. 2019. Influence of vegetation coverage and climate environment on soil organic carbon in the Qilian mountains. Scientific Reports, 9(1), 17623. doi: 10.1038/s41598-019-53837-4

**17.  L102: "Samples of…were collected…"**

Response: We have revised this sentence according to your suggestion. The revised sentence is as follows:

Samples of precipitation, groundwater, soil, and plant were collected at Lenglong (alpine meadow), Hulin (forest), and Xiying (arid foothills) in the Shiyang River Basin from April 2018 to October 2019 (Table 1)

**18.  L109: "telling the date"?**

Response: We have changed this sentence to" Simultaneously, the polyethylene bottle sample is labeled with the date and type of precipitation (rain, snow, hail and rain)".

**19.  L111: Are there any replicates for soil samples of each soil layer?**

Response:    Yes, it is. Two soil samples were collected for each soil layer. Part of the soil samples were dried to measure the soil moisture content, and the other part was used to conduct isotope experiments to obtain soil water isotope values.Each sample was measured 6 times, and the average value was taken to obtain the soil water isotope value.

**20. L117: How many plant species are sampled? How about the position of sampled stems in the canopy? What is the size of stem samples? "xylem stem" should be "stem".**

Response: We collected 3 planting quilts in total, Qinghai spruce and purple-winged salsola in Lenglong, Qinghai spruce in Hulin , and poplar in Xiying. Our sampled stems are located at the bottom right of the tree canopy, which means we are collecting the oblique branches of the tree. The sample size of Qinghai spruce and poplar we collected is about 50cm, and the sample size of herbaceous plants such as Salsola is about 10cm. We have changed "xylem stem" to "stem".

**21. L120: How is the groundwater sampled? What is the depth of water table at each sampling point.**

Response: Groundwater samples were obtained from the groundwater monitoring wells of the Shiyang River Basin Administration, China Hydrological Administration and Gansu Hydrological Administration. The sampling interval is monthly. The depth of groundwater in alpine meadows is 30-60 m, the depth of groundwater in forests is 15-30 m, and the depth of groundwater in arid foothills is 2.5-15 m.

**22. L126: How many isotope standards were used?**

Response: We only used one isotope standard, SMOW, which is the standard mean ocean water, as a unified standard for the isotopes of hydrogen and oxygen.

**23. L133: "Due to the existence of methanol and ethnol in plant water samples…"**

Response: We have revised this paragraph according to your suggestion. The revised sentence is as follows:

Due to the existence of methanol and ethnol in plant water samples, it is necessary to modify plant samples' original data.

**24. L148: Since different water lines have been defined here, I suggest the revised manuscript used their abbreviations hereafter.**

Response: We have used abbreviations for the different water lines in the revised manuscript.

**25. L148-150: These information should be mentioned in the Introduction section.**

Response: We have added these information in the Introduction section

**26. L152-159: Sentences are repeated here.**

Response: We have deleted the repeated sentences.

**27. L163: Is there any data or references for such statements.**

Response: We added relevant references and rewritten this sentence:

According to the Natural Resources Survey Report of Shiyang River Basin in 2020, the vegetation coverage rate of the alpine meadow is 25.95%, and the vegetation coverage rate of the arid foothills is 8.48%. The vegetation coverage rate of the alpine meadow is higher than that of the arid foothills, with better water retention ability and less evaporation of soil moisture (Wan et al., 2019; Wei et al., 2019).

Wan, Q. Z.,  Zhu, G. F., Guo, H. W., Zhang, Y., Pan, H. X ., and Yong, L. L et al. 2019. Influence of vegetation coverage and climate environment on soil organic carbon in the qilian mountains. Scientific Reports, 9(1), 17623. doi: 10.1038/s41598-019-53837-4

Wei, W., Xie, B., Zhang, X., and Zhang, J. 2019. Spatial heterogeneity of soil moisture and vegetation cover in Shiyang river basin, northwest china. IOP Conference Series: Earth and Environmental Science, 237(5), 052003 (5pp). doi: 10.1088/1755-1315/237/5/052003

**28. L169-171: These results should be based on proper statical analysis.**

Response: The revised sentence is as follows:

According to the weighted average of stable isotopes of various water bodies (Table 2), the soil water isotope value of alpine meadows is -9.16‰, which is the most depleted and the closest to the precipitation isotope value (-9.44‰).

**29. L190-192: Any data or references?**

Response: We have added relevant   references:

Csilla, F., Györgyi, G., Zsófia, B., and Eszter, T. 2014. Impact of expected climate change on soil water regime under different vegetation conditions. Biologia(11), doi:10.2478/s11756-014-0463-8.

Li,L. F., Yan, J.P., Liu, D. M., Chen, F., and Ding, J. M.2009. Changes in soil water content under different vegetation conditions in arid-semi-arid areas and analysis of vegetation construction methods. Bulletin of Soil and Water Conservation, 29(001), 18-22.

Western, A. W., and Grayson, R. B. 1998. The tarrawarra data set: soil moisture patterns, soil characteristics, and hydrological flux measurements. Water Resources Research, 34(10), 2765-2768. doi: 10.1029/98WR01833.

**30. L192-194: References are also needed here.**

Response: The increase of vegetation coverage rate of the alpine meadow is calculated according to the rise in grassland area, with data from the comprehensive natural survey report of Shiyang River Basin. According to the complete survey report on the natural resources of the Shiyang River Basin, the natural grassland area of the

Shiyang River Basin was 12,452.72 hectares in 2017 and 13,071.94 hectares in 2019. We can see that from 2017 to 2019, the grassland area of the Shiyang River Basin increased by 619.22 hectares, which shows the increase in vegetation coverage in the Shiyang River Basin. In addition, we have added a description of the mechanism explanation :

On the one hand, with the increase of vegetation restoration, the area of natural grassland in the Shiyang River Basin has increased. Alpine meadows account for the most significant proportion in the Shiyang River Basin, which increases the soil's water retention capacity in the alpine meadows and reduces the amount of soil water evaporation. On the other hand, there is a lot of precipitation in the upper reaches of the Shiyang River. According to Table 1, Lenglong, a representative of alpine meadows, has an average annual precipitation of 595.10 mm, a low temperature, and an average annual temperature of -0.20°C. The lower temperature and higher precipitation also make the soil water evaporation intensity weak in the alpine meadow.

**31. L192-198: Since this is the Results section, I suggest move these content to Discussion.**

Response: We have moved this part to the discussion section.

**32. L197: "The dry foothills…".**

Response: We have changed "The dry and dry foothills" to "The dry foothills."

**33. L209, L212: "affluent" and "abundant" are not proper words here.**

Response: We have deleted "affluent" and "abundant" and replaced it with "enrichment".

**34. L216-217: This is not a convincing conclusion.**

Response: We checked the relevant literature and corrected this conclusion:

In the dry season, alpine meadow plants have the highest concentration of water isotopes (-2.84‰). There is no overlap between soil water and plant water, indicating that alpine meadow plants do not directly use soil water in the dry season. Plant water isotope (-6.04‰) and precipitation isotope value (-6.40‰) in the rainy season are close. The surface and deep layers of groundwater and soil water intersect, indicating

that plant water in the alpine meadow in the rainy season are mainly supplied by precipitation. Groundwater does not directly use precipitation, but Rely on soil water for replenishment. In the dry season, due to the low temperature (average temperature of 0.30°C), there is a large amount of melting ice and snow in alpine meadows, abundant precipitation and abundant melting water, and plants do not directly use soil water. In the rainy season (average temperature 8.72°C), as the temperature rises, plant water isotopes undergo intense evaporative fractionation, and isotopes are enriched. With the increase of precipitation, the surface runoff increases, and the soil underwater infiltrates the groundwater.

**35. L244: Why use 8℃ turning point?**

Response: If the temperature is below 0°C, the air will expand adiabaticly and the water vapor will change adiabatic cooling (Rozanski, 1992). When the temperature is between 0°C and 8°C, the influence of local water vapor circulation is greater. When the temperature is below 8°C, the secondary evaporation under the clouds is very strong (Ma et al., 2018). Therefore, the temperature is divided into three gradients (below 0°C, between 0°C and 8°C and above 8°C) to analyze the relationship between precipitation isotope and temperature. So we use 8℃ as the turning point.

**36. Fig.1: "Shiyang River system"? Is it "Shiyang River Basin"? The letters (a, b, and c) should be explained in figure caption.**

Response: The Shiyang River system is not the Shiyang River Basin. The Shiyang River system refers to the tributaries and main streams of the Shiyang River Basin. We have added the entire Shiyang River Basin map to Fig. 1.

[Figure]

Figure. 1 Study area and observation system

**37. Fig.2: Please rearrange the graphs in a single column or row.**

Response: We have rearranged the graph into a single row according to your suggestion.

[Figure]

**38. Fig.3: "θ" should be defined here. I also suggest the graphs on the left and right panel using a same x-axis range, respectively.**

Response: We have defined θ in the figure according to your suggestions, and used the same x-axis range to draw the graphs on the left and right panels.

[Figure]

**39. Fig.4: Where are the standard deviations or errors?**

Response: We have added error bars to the graph.

[Figure]

**40. Fig.6: Please define M1, M2 and M3 in the figure caption. How is the situation of hydrogen isotope?**

Response: We have defined $M_1$, $M_2$, and $M_3$ in the caption of Fig. 6. In Section 4.2, a discussion on the altitude effect of hydrogen isotopes is added:

The study area is divided into the rainy season (May-September) and dry season (10-April of the following year), and the relationship between altitude and isotope is analyzed (Fig. 7). The altitude effect of precipitation isotope is stronger than the relationship between soil water isotope and altitude and the relationship between plant

water isotope and altitude, but the relationship between plant water $\delta D$ and altitude in the rainy season is stronger than the relationship between soil water $\delta D$ and altitude. It shows that in SPAC, precipitation isotope is most affected by altitude, and plant water isotope is least affected by altitude. As the quality of water vapor rises along the hillside, the temperature continues to decrease, and the isotopic values of precipitation continue to be consumed. From the arid foothills to alpine meadows, the elevation rises from 2097m to 3647m. The average values of precipitation isotopes $\delta^{18}O$ and $\delta D$ changed from -7.33‰ to -9.10‰, and from -48.62‰ to -54.93‰, respectively. The rate of change was -0.11‰$(100m)^{-1}$, -0.41‰$(100m)^{-1}$ , In the globally recognized precipitation $\delta18O$ altitude gradient range, this rate of change is -0.28‰ $(100m)^{-1}$ (Porch and Chamberlain, 2001). The squares of correlation coefficients between $\delta^{18}O$ and $\delta D$ of rainy season precipitation and altitude are 0.79 and 0.98. The rate of change is -0.12‰$(100m)^{-1}$ and -1.05‰$(100m)^{-1}$, respectively. In the dry season, the correlation coefficient squares of $\delta^{18}O$ and $\delta D$ with altitude are 0.88 and 0.90, respectively, and the rate of change is -0.18‰$(100m)^{-1}$ and -0.79‰$(100m)^{-1}$, respectively. It can be seen that the altitude effect of precipitation $\delta^{18}O$ is stronger in the dry season ($R^2$=0.88) than in the rainy season ($R^2$=0.79), and the altitude effect of precipitation $\delta D$ is stronger in the rainy season ($R^2$=0.98) than in the dry season ($R^2$=0.90). The relationship between soil water isotope and altitude is stronger in the rainy season ($R^2$=0.26, $R^2$=0.73) than in the dry season ($R^2$=0.28, $R^2$=0.26). The relationship between plant water $\delta^{18}O$ and altitude is stronger in the dry season ($R^2$=0.11) than in the rainy season ($R^2$=0.11), and the relationship between plant water

δD and altitude in the rainy season ($R^2=0.62$) is stronger than that in the dry season ($R^2=0.56$). It can also be seen from the figure that there are anti-elevation shows in some areas, mainly from forests to dry foothills. This may be related to the existence of reservoirs in the arid foothills. Reservoirs may cause the reversal of the local water vapor cycle-the anti-elevation effect. Generally speaking, there is a negative correlation between altitude and SPAC isotope composition. The altitude effect of precipitation isotope is stronger than the relationship between soil water isotope and altitude, and stronger than the relationship between plant water isotope and altitude.

[Figure]

Fig. 7 Relationship between different isotope and altitude in the dry season (a, c ) and in the rain season (b, d), $M_1$ stands for alpine meadows, $M_2$ stands for forests, and $M_3$ stands for arid foothills

**41. Table 1: Are the comparisons including significance testing?**

Response: Because Table 1 is the basic parameter information of the sampling points of the article, including the latitude and longitude, altitude, average annual temperature and average annual precipitation of each sampling point, all the comparisons in Table 1 have not been tested for significance. In Table 2 and Table 3, the comparison of the data has been tested for significance.

---

## Author Response (AR2)

Dear Editor and Reviewers:

Thank you for your letter and for the reviewer's comments concerning our manuscript entitled "Difference of SPAC composition and control factors of different vegetation zones in north slope of Qilian Mountains" (Manuscript Number: bg-2021-127).

Based on the reviewers' comments, we carefully checked the writing style in Introduction and Discussion, and carefully revised our manuscript. The revised parts are marked in red in the revised version of the manuscript. The main corrections and responses to the reviewers' comments are as follows.

**Responses to the reviewer's comments:**

**Editor**

**Comment: Kindly incorporate the attached comments and submit your revised manuscript along with point by point response wherever required.**

Response: We have carefully reviewed our manuscript based on your suggestions and those of other reviewers, and we have worked hard to improve the presentation quality of our manuscript. The revised content has been marked in red font of the manuscript.

**Response to Reviewer #1**

**Reviewer #1: In the current revision, Liu et al. have addressed most of my concerns in last referee report. Particularly, they reorganized the structure of the whole manuscript and raised specific scientific question as suggested before. Nevertheless, I am still not satisfied with the manner of writing of Introduction**

**and Discussion, especially their language issues which were mentioned in last report. In Introduction, the first two paragraphs employed many references to underline the water cycle of SPAC and the advantages of stable isotopes. I suggest the authors to focus the scientific question and make it more specific. In Discussion, e.g., 4.22, the authors presented mainly the complete results, but they didn't try to generalize the drivers and mechanisms behind the data, and no previous studies were listed and compared with each other. In addition, I still have some specific comments.**

Response: We have revised the Introduction and Discussion based on your suggestions. The revised content is as follows:

**1 Introduction**

[revised manuscript text omitted]

**Specific comments**

**1. L14: Replace "studied" with "investigated" or other words.**

Response: We have revised this sentence based on your suggestion. The revised sentence is as follows:

In this study, we investigated the changes of stable water isotopes in the SPAC in three different vegetation zones (alpine meadow, forest, and arid foothills) in the Shiyang River Basin.

**2. L15: Delete "soil-plant-atmosphere continuum".**

Response: We have revised this sentence based on your suggestion. The revised sentence is as follows:

In this study, we investigated the changes of stable water isotopes in the SPAC in three different vegetation zones (alpine meadow, forest, and arid foothills) in the Shiyang River Basin.

**3. L28: Delete "in water technology".**

Response: We have revised this sentence based on your suggestion. The revised sentence is as follows:

The relative abundance changes of oxygen and hydrogen isotopes in water can indicate the water cycle and the water use mechanism in plants, so isotope technology has become an increasingly important method for studying the water cycle (Gao et al., 2009; Song et al., 2002; Coplen, 2013; Shou et al., 2013).

**4. L41-44: I suggest the authors to modify this sentence as "As an effective tool, stable isotope technology is widely applied in studying…, and tracing the way plants use water…".**

Response: We have revised this sentence based on your suggestion. The revised sentence is as follows:

As an effective tool, stable isotope technology is widely applied in studying the relationship between environmental factors and the water cycle (Araguás-Araguás et al., 1998; Christopher et al., 2009), water transportation, and distribution mechanisms (Gao et al., 2011), and ways of tracing water use by plants (Detjen et al., 2015).

**5. L47: Ehleringer.**

Response: We have revised this sentence based on your suggestion. The revised sentence is as follows:

The understanding of the relationship between the influence of plant characteristics, water use efficiency and water sources (Ehleringer, 1991; Sun et al., 2005; Li et al., 2019) provides a new observation method for revealing the water cycle mechanism of the hydrological ecosystem (Nie et al., 2014; Yu et al., 2007; Wang et al., 2019).

**6. L49: Delete "stable", "methods".**

Response: We have revised this sentence based on your suggestion. The revised sentence is as follows:

Hydrogen and oxygen isotope have been used to study the water cycle at the interface of "soil-root," "soil-plant," and "soil-atmosphere," but only a small number of parameters play an important role in the complex interactions of various surfaces (Durand et al., 2007; Li et al., 2006; West et al., 2010).

**7. L51: Modify this sentence to be specific, what the parameters are.**

Response: We have revised this sentence based on your suggestion. The revised sentence is as follows:

Hydrogen and oxygen isotopes have been used to study the water cycle at the interface of "soil-root", "soil-plant", and "soil-atmosphere", but only a few parameters play an important role in the complex interactions between various surfaces (Durand et al., 2007; Li et al., 2006; West et al., 2010). Previous studies have shown that local factors, especially temperature, mainly control stable isotope precipitation changes in mid-latitudes (Dai et al., 2020).

**8. L63: Delete "of water".**

Response: We have revised this sentence based on your suggestion. The revised sentence is as follows:

Through the research on the composition of hydrogen and oxygen isotopes in different water bodies, we can further understand the mechanism of water use by vegetation (Yang et al., 2015) and provide a scientific basis for vegetation restoration

in arid and semi-arid areas.

**9. L61-66: Please rearrange this long sentence, especially the content after "such as". "…in western China"?**

Response: We have condensed this sentence based on your suggestion.. The revised sentence is as follows:

Through the research on the composition of hydrogen and oxygen isotopes in different water bodies, we can further understand the mechanism of water use by vegetation (Yang et al., 2015) and provide a scientific basis for vegetation restoration in arid and semi-arid areas.

**10. L67: "…extend the results of the small-scale…".**

Response: We have revised this sentence based on your suggestion. The revised sentence is as follows:

In the existing research, how to extend the results of the small-scale SPAC water cycle research to the large-scale area has become a hot spot and difficulty.

**11. L68: Delete "in the current research".**

Response: We have revised this sentence based on your suggestion. The revised sentence is as follows:

In the existing research, how to extend the results of the small-scale SPAC water cycle research to the large-scale area has become a hot spot and difficulty.

**12. L69: I think that the stable isotope technology is not much new in studying SPAC.**

Response: We have removed this content from the manuscript based on your

suggestion.

**13. L74-76: Although the isotope ratio in soil water changes with depth, it remains stable when transporting from plant roots to stems, leaves or young unbolted branches.**

Response: We have revised this sentence based on your suggestion. The revised sentence is as follows:

Although the isotope ratio in soil water varies with depth, it remains stable when transferred from plant roots to stems, leaves or young unbolted branches (Porporato, 2001; Meissne et al., 2014).

**14. L89-90: "…runoff regeneration, groundwater recharge and regeneration capacity can be determined".**

Response: We have revised this sentence based on your suggestion. The revised sentence is as follows:

Combined the isotopic composition changes of surface water, soil water and groundwater, precipitation infiltration and runoff generation process (Bam and Ireso, 2018; Hou et al., 2008), groundwater recharge and regeneration capacity (Smith et al., 1992; Cortes and Farvolden, 1989) can be determined.

**15. L93: What is "it"? "…studying the mechanism of the deep water cycle".**

Response: We modified this sentence based on the context. The revised sentence is as follows:

Regional meteorological and hydrological conditions and the contribution of various environmental factors can be evaluated (Hua et al., 2019) by comparing

different waterline equations and analyzing changes in various water bodies.

**16. L94: "…plants transpiration".**

Response: We have revised this sentence based on your suggestion. The revised sentence is as follows:

As an important component of the global water cycle, plants control 50-90% of transpiration (Jasechko et al., 2013; Coenders-Gerrits et al., 2014; Schlesinger and Jasechko, 2014).

**17. L125: "…better than that of lower reaches…".**

Response: We have revised this sentence based on your suggestion. The revised sentence is as follows:

The vegetation coverage in the upper and middle alpine regions is better than that of the lower reaches, with trees, shrubs, and grass covered (Wan et al., 2019).

**18. L148: "xylem stem" should be "stem".**

Response: We have revised this sentence based on your suggestion. The revised sentence is as follows:

The vegetation samples are collected with a sampling shear. First, we peel off the bark and put the stem into a 50 ml glass bottle. After that, we sealed the bottle mouth and keep it frozen before the experimental analysis.

**19. L158: "…and used…".**

Response: We have revised this sentence based on your suggestion. The revised sentence is as follows:

To eliminate the memory effect of the analyzer, we discarded the values of the first two injections and used the average of the last four injections as the final result

value.

**20. L159-162: Please standardize the font of "δ" throughout the manuscript.**

Response: We have unified the font of "δ" in the manuscript to New Rome according to your suggestion, and the revised content has been marked in red in the manuscript.

**21. L163: "modify" to "calibrate".**

Response: We have revised this sentence based on your suggestion. The revised sentence is as follows:

Due to the existence of methanol and ethanol in plant water samples, it is necessary to calibrate the original data of plant samples.

**22. L175: "…normally distributed…"?**

Response: We have revised this sentence based on your suggestion. The revised sentence is as follows:

Since the isotopic data are generally normally distributed according to the Kolmogorov-Smirnov (KS) test.

**23. L202: "The slope of LMWL…".**

Response: We have revised this sentence based on your suggestion. The revised sentence is as follows:

The slope of LMWL of alpine meadows (7.88), forests (7.82), and arid foothills (7.72) are all smaller than that of GMWL(8.00).

**24. L215-224: As I understand, the current paragraph is only about oxygen isotope. Is it right? So it should be "stable oxygen isotope" since L215.**

Response: Yes, we are introducing oxygen isotopes since L215. We have revised this sentence based on your suggestion. The revised sentence is as follows:

According to the weighted average of stable oxygen isotopes of various water bodies (Table 2), alpine meadows' soil water $\delta^{18}O$ is -9.16‰, the most depleted and the closest to the precipitation $\delta^{18}O$ (-9.44‰).

**25. L216: I suggest the authors use $\delta^{18}O$ and $\delta D$ in manuscript instead of "water isotope value".**

Response: We have revised this sentence based on your suggestion. The revised sentence is as follows:

According to the weighted average of stable oxygen isotopes of various water bodies (Table 2), alpine meadows' soil water $\delta^{18}O$ is -9.16‰, the most depleted and the closest to the precipitation $\delta^{18}O$ (-9.44‰).

**26. L227: "…plant water isotope…".**

Response: We have revised this sentence based on your suggestion. The revised sentence is as follows:

**3.3 Relationship between soil water and plant water isotope in different vegetation zone**

**27. L238: "…the current experiment is divided into…".**

Response: We have revised this sentence based on your suggestion. The revised sentence is as follows:

According to the study area's precipitation, the current experiment is divided into dry season (October-April of the following year) and the rainy season (May-September) for analysis (Fig. 4).

**28. L240: "…have the highest value of δ18O (-2.84‰)".**

Response: We have revised this sentence based on your suggestion. The revised sentence is as follows:

In the dry season, alpine meadow plants have the highest value of $\delta^{18}O$ (-2.84‰).

**29. L251: What's the meaning of "soil underwater infiltrates the groundwater"?**

Response: We originally wanted to express in this sentence that as the temperature rises during the rainy season (the average temperature is 8.72°C), plant water isotopes undergo intense evaporative fractionation and isotope enrichment. As precipitation increases, surface runoff increases, and water seeps from the soil into the groundwater. We have modified this sentence. The revised sentence is as follows:

As the increase of temperature (average temperature 8.72°C), precipitation and surface runoff increases, and water infiltrate into groundwater from soil.

**30. L259: "…plant water oxygen is the most enriched…".**

Response: We have revised this sentence based on your suggestion. The revised sentence is as follows:

In the dry season, plant water oxygen is the most enriched, and the isotopic values of groundwater and soil water are close.

**31. L262: "meters" to "m". "burial" to "table".**

Response: We have revised this sentence based on your suggestion. The revised sentence is as follows:

According to the natural resources survey report of the Shiyang River Basin, the buried groundwater level in the arid foothills is 2.5-15 m, and the groundwater table is relatively shallow, making the soil water in the arid foothills mainly recharged by groundwater in the dry season.

**32. L266: "…soil water isotope".**

Response: We have revised this sentence based on your suggestion. The revised sentence is as follows:

**4.1 Variation of soil water isotope and SWC between different vegetation zone**

**33. L270: "… (-0.15), while that of the forest…".**

Response: We have revised this sentence based on your suggestion. The revised sentence is as follows:

The coefficient of variation of the arid foothills is the largest (-0.15), while that of the forest is the smallest (-0.25), indicating that from forest to arid foothills, the closer to arid regions, the greater the coefficient of variation and that the greater the instability of stable isotope soil water.

**34. L283: "soil".**

Response: We have revised this sentence based on your suggestion. The revised sentence is as follows:

Alpine meadows account for the most significant proportion in the Shiyang River Basin, which increases the soil water retention capacity in the alpine meadows and reduces the amount of soil water evaporation.

**35. Fig. 2: Relative humidity.**

Response: We have modified Fig. 2 according to your suggestion, and the revised Fig.2 is as follows:

[Figure]

Fig. 2 Diurnal variation of relative humidity, precipitation, temperature, and swc (%)
from April 2018 to October 2019

**36. Fig. 4 and Fig. 5: Add "(m)" after "Soil depth" and delete "m" along the
Y-axis.**

Response: We have modified Fig.4 and Fig. 5 according to your suggestion, and the
revised Fig. 4 and Fig. 5 are as follows:

[Figure]

Fig. 4 (a)-(c) represents the variation of δ¹⁸O of soil, plant, precipitation and groundwater with soil depth in the alpine meadow, forests and arid foothills in the dry season, and (a₁)-(d₁) represents the variation of δ¹⁸O of soil, plant, precipitation and groundwater in the alpine meadow, forests and arid foothills in the rainy season

[Figure]

Fig.5 The variation of $\delta^{18}O$ and soil water content ($\theta$, %) with soil depth. (a)-(c) represent alpine meadow, forests and arid foothills, respectively

**36. Fig. 8: Please standardize the font size. "absoult" should be "absolute"?**

Response: We have modified Fig. 8 according to your suggestion, and the revised

Fig.8 is as follows:

[Figure]

Fig. 8 Relationship between different isotope and relative humidity and precipitation,

$M_1$ stands for alpine meadows, $M_2$ stands for forests, and $M_3$ stands for arid foothills

---

## Author Response (AR3)

Dear Editor:

Thank you for your letter and your comments concerning our manuscript entitled "Difference of SPAC composition and control factors of different vegetation zones in the north slope of Qilian Mountains" (Manuscript Number: bg-2021-127).

Based on your comments, we carefully checked the grammar and revised our manuscript. The revised parts are marked in red in the revised version of the manuscript.

**Responses to the Editor's comments:**

**Comment:** Thanks for revising the article. However, I could find grammatical mistakes at several places in the manuscript. I would strongly recommend you to get the article checked for English and submit the corrected version.

**Response:** We have carefully reviewed our manuscript based on your suggestions, and we have worked hard to get the article checked for grammar. The revised content has been marked in red font of the manuscript.